# Regulation of RNA Splicing: Aberrant Splicing Regulation and Therapeutic Targets in Cancer

**DOI:** 10.3390/cells10040923

**Published:** 2021-04-16

**Authors:** Koji Kitamura, Keisuke Nimura

**Affiliations:** 1Division of Gene Therapy Science, Department of Genome Biology, Graduate School of Medicine, Osaka University, 2-2 Yamada-oka, Suita, Osaka 565-0871, Japan; kkitamura@ent.med.osaka-u.ac.jp; 2Department of Otorhinolaryngology-Head and Neck Surgery, Graduate School of Medicine, Osaka University, 2-2 Yamada-oka, Suita, Osaka 565-0871, Japan

**Keywords:** RNA splicing, aberrant splicing, cancer, splicing factor, splicing variant, non-coding RNA, treatment targeting splicing

## Abstract

RNA splicing is a critical step in the maturation of precursor mRNA (pre-mRNA) by removing introns and exons. The combination of inclusion and exclusion of introns and exons in pre-mRNA can generate vast diversity in mature mRNA from a limited number of genes. Cancer cells acquire cancer-specific mechanisms through aberrant splicing regulation to acquire resistance to treatment and to promote malignancy. Splicing regulation involves many factors, such as proteins, non-coding RNAs, and DNA sequences at many steps. Thus, the dysregulation of splicing is caused by many factors, including mutations in RNA splicing factors, aberrant expression levels of RNA splicing factors, small nuclear ribonucleoproteins biogenesis, mutations in snRNA, or genomic sequences that are involved in the regulation of splicing, such as 5’ and 3’ splice sites, branch point site, splicing enhancer/silencer, and changes in the chromatin status that affect the splicing profile. This review focuses on the dysregulation of RNA splicing related to cancer and the associated therapeutic methods.

## 1. Introduction

Previous studies have revealed that RNA splicing is essential for mRNA maturation and regulation of gene expression. Alternative splicing generates multiple mRNAs from a limited number of genes, leading to the synthesis of a variety of proteins and functional diversity. This process plays an important role in cell differentiation and development. RNA splicing is regulated by a number of factors that determine gene expression and function, and is closely related to the transcriptional machinery involved in RNA polymerase (RNAP) II, translational processes, and epigenetic processes associated with chromatin status. In cancer, some specific splicing variants have been shown to contribute to cancer development, progression, and resistance to treatment. Cancer-related mis-splicing affects not only gene expression levels but also the transcriptional machinery, epigenome, and downstream signaling pathways for cancer survival. Therefore, understanding aberrant splicing mechanisms in cancer would help identify cancer biomarkers and develop new therapeutic strategies. This review discusses recent studies on aberrant alternative splicing associated with cancer and the targeted therapeutic methods.

## 2. Splicing Mechanism

Splicing is a critical process for the removal of introns and long non-coding RNAs from precursor mRNA (pre-mRNAs) to generate mature RNAs, and many genes are differentially expressed through splicing-related regulation. Alternative splicing also leads to the transcription of a single gene into multiple and different mature mRNAs. This mechanism enables the production of various proteins from a limited number of genes.

Splicing is a catalytic process caused by dynamic changes in large protein-RNA complexes called spliceosomes. Spliceosomes are classified into major and minor spliceosomes. The major spliceosome is a complex containing five small nuclear ribonucleoproteins (snRNPs) and more than 170 proteins, which enables it to recognize most splice sites (ss) and is involved in the removal of more than 99% of introns [1]. The U2-type intron has the sequence GU-AG at the 5′ and 3′ splice sites, whereas the U12-type intron has the sequence AU-AC. Major spliceosomes mainly splice U2-type introns, whereas minor spliceosomes splice U12-type introns.

The major spliceosome comprises U1, U2, U4/U6, and U5 snRNPs. Its assembly starts when U1 snRNP binds to the 5′ ss on the pre-mRNA, and U2 snRNP binds to the branch point site (BPS) with the aid of the U2 auxiliary factor (U2AF). U4 and U6 snRNPs are usually polymerized by base pairing between RNAs and, together with U5 snRNP, then associate with the U1/pre-mRNA/U2 complex to form a functional spliceosome. In contrast, the minor spliceosome comprises U5, U11, U12, U4atac, and U6atac. U11 and U12 snRNPs recognize the 5′ ss and BPS, respectively. Only the U5 snRNP is found in both spliceosome types, whereas U4atac and U6atac snRNPs play the same role in the minor spliceosome as the U4/U6 nsRNP heterodimer complex in the major spliceosome.

These alternative splicing events are regulated by both cis-regulatory and trans-acting factors. RNA sequence analysis has revealed that specific RNA regions such as ss sequences, BPS, exonic splicing enhancer (ESE), exonic splicing suppressor (ESS), intronic splicing enhancer (ISE), and intronic splicing suppressor (ISS), which are cis-regulatory elements, are recognized by the spliceosome and are thus involved in splicing control. In addition, the SR protein and heterogeneous nuclear ribonucleoproteins (hnRNPs), which are trans-acting factors and splicing factors, recognize and bind to the enhancer (ESE, ESS) or silencer (ISE, ISS) motifs of pre-mRNA, thereby regulating the binding of spliceosome to pre-mRNA and controlling splicing [2,3]. In addition to SR proteins and hnRNPs, numerous other RNA-binding proteins have been reported to regulate alternative splicing, including CELF [4], MBNL [5], and NOVA [6].

## 3. Splicing Is Regulated by Non-Coding RNAs Including snRNA and sno/scaRNA

snRNPs form spliceosomes with snRNAs, non-coding RNAs, and components of the spliceosome, and various proteins. snRNPs affect splicing efficiency and ss selection by binding to pre-mRNA and by interacting with spliceosomes. Splicing is regulated by the abundance and biogenesis levels of snRNP and by the abundance of and mutations in snRNAs. Survival of motor neuron (SMN) and a subset of GEMIN proteins are essential for snRNP biogenesis. These proteins associate with snRNAs to form SMN complexes that are required for the interaction between Sm proteins and snRNA during snRNP assembly. The deficiency in SMN protein in spinal muscular atrophy (SMA) decreases tissue-specific snRNP biogenesis levels, resulting in the alteration of alternative splicing [7,8]. Other studies also support that alternative splicing is regulated by SMN complex components, the GEMIN2 protein, and SmB/B’ protein that make up the Sm ring, through changes in snRNP biogenesis levels [9,10].

Post-transcriptional modifications of snRNAs are involved in the regulation of splicing patterns. In eukaryotic cells, 2′-O-methylation and pseudouridine (Ψ) are two major post-transcriptional modifications of snRNA, although N6-methyladenosine (m^6^A) RNA methylation is a modification of mRNA. Small RNAs, as well as proteins, carry out these post-transcriptional modifications. Small nucleolar RNAs (snoRNAs) are non-coding RNAs that act as guides for guiding RNA-dependent RNA modifications. These guide RNAs act as catalysts through complementary binding to the target RNA. These guide RNAs are located mainly in the nucleolus (snoRNAs) and nucleoplasmic Cajal bodies (scaRNAs). Sno/sca RNAs are classified into box C/D and box H/ACA, combined with RNA binding proteins containing methyltransferase and pseudouridine synthase, respectively, to form sno/scaRNPs to function as guides. snRNAs are modified by box C/D snoRNAs for 2′-O-methylation and by box H/ACA for pseudouridine, and play an important role in RNA–RNA interaction in the spliceosome and splicing fidelity [11]. The extent and location of these modifications change in response to intracellular stress [12]. In cancer, the type and abundance of sno/scaRNA differ according to cancer type [13]. Sno/scaRNAs have also been reported to have tumor-suppressive or oncogenic functions, depending on the type of cancer, and to be associated with cellular processes such as tumor growth, invasion, metastasis, cell death, angiogenesis, and cancer-related signaling pathways [14]. Differential expression of sno/scaRNAs has been associated with patient prognosis [15]. For example, in hepatocellular carcinoma, low expression of snoRNA increases lipid accumulation and is associated with poor prognosis. Furthermore, Cajal bodies have various functions related to the processing, modification, and maturation of snRNPs and are observed only in a limited number of human cells, such as neurons and cancer cells. The depletion of the Cajal body causes aberrant splicing by suppressing the transcription of sn/snoRNA and limiting the biogenesis levels of spliceosomal snRNP through snRNA modification and changes in their turnover [16,17]. These results suggest that the modulated expression of guide sno/scaRNAs can be linked to snRNA activation in cancer. Further studies are needed to identify sno/scaRNPs as biomarkers and therapeutic targets for cancer.

Recent studies have shown frequent A>C somatic mutations at the third base of U1 snRNA in several types of tumors, such as chronic lymphocytic leukemia (CLL) and hepatocellular carcinoma, and frequent A>G mutations at the third base in approximately 50% of Sonic hedgehog medulloblastomas. The main function of U1 snRNA is to recognize the 5′ ss, resulting in base pairing. The mutated U1 snRNA recognizes the ectopic 5′ ss, leading to the creation of new splicing junctions that alter the splicing patterns of many genes, including known cancer driver genes, and promotes tumorigenesis [18,19] (Figure 1, Table 1). SnRNAs are equally present in the spliceosome, but their relative levels vary greatly between tissues and between cancer samples, and also affect splicing in cancer. In breast cancer, genes that are particularly sensitive to changes in snRNA abundance in cell lines were found to be similarly preferentially mis-spliced in clinical diverse cohorts of invasive breast ductal carcinomas. Perturbation of individual snRNAs extensively promoted gene-specific differences in alternative splicing, but not in global transcriptomic splicing [20]. Another study using antisense morpholino oligonucleotides for U1 snRNA to decrease snRNA abundance found that low doses of U1 antisense morpholino oligonucleotides increased migration and invasion in cancer cells and were associated with oncogene upregulation and tumor suppressor gene downregulation, whereas U1 snRNA overexpression showed the opposite phenotype. These results indicate that the change in U1 expression level affects the expression levels of oncogenes and oncogenic phenotypes [21].

As aberrant mRNA splicing is seen in many cancers, further studies of snRNPs and snRNAs, as well as splicing factors, are required to fully understand such regulated pre-mRNA processing.

## 4. Splicing Interacts with Histone Modification in Tumorigenesis

Splicing is temporally and spatially linked to transcriptional mechanisms and chromatin structures. Transcription kinetics and histone marks are involved in the decision process of whether the target exons and introns are included or excluded [22]. In the transcription kinetics function on splicing, the transcriptional elongation rate of RNAPII affects alternative splicing by influencing the detection rate of splice sites and splicing regulatory sequences in pre-mRNA. DNA sequences that cause RNAPII pausing and inhibition of RNAPII elongation promote the inclusion of exons into mature RNA [23,24]. Drugs that enhance chromatin accessibility and factors promoting elongation also increase exon skipping [25,26,27]. For example, CTCF, a DNA-binding protein involved in insulators, promotes the inclusion of exon5 of CD45 by binding to the target site in exon5. However, DNA methylation of exon5 inhibits CTCF binding, promotes RNAPII elongation, and induces exon5 skipping [28,29]. The effect on the transcription elongation rate may depend not only on specific promoters that stimulate transcription, but also on specific splicing factors. SRSF2 stimulates transcriptional elongation and processing of a subset of genes [30] and recruits the positive transcription elongation factor b (P-TEFb) by binding to specific sites in nascent RNA, thus releasing RNAPII from promoter-proximal pauses [31]. Thus, an appropriate elongation rate is essential for the selection of alternative splicing exons. It has also been shown that RNAPII transcription pausing is required in yeast for co-transcriptional splicing [32], suggesting that co-transcriptional splicing is an evolutionarily conserved mechanism.

In the histone modification function on splicing, genome-wide analysis revealed that inhibition of HDAC associated with histone acetylation altered the splicing pattern of 700 aberrant genes and reduced the co-transcription of SRSF5 with its target exons [33]. Histone modifications also recruit splicing factors through chromatin-binding proteins, suggesting that histone marks may also influence splicing [34,35]. H3K36me3 is a histone modification that is highly enriched near transcriptionally active genes or intron–exon boundaries [36,37]. H3K36me3 recruits MRG15, a chromodomain-containing protein that is part of the histone modification complex, which in turn recruits the splicing factor PTB. PTB binds to pre-mRNA-specific sequences and causes alternative splicing depending on the position of the binding site [35,38]. The H3K36me3-MRG15-PTB complex forms a chromatin co-transcriptional splicing structure that retains the splicing mechanism near the chromatin structure. SETD2 has H3K36me3 enzymatic activity [39]. In colorectal cancer, although SETD2 is not required for homeostasis in the intestine, SETD2 depletion results in tumorigenesis in the intestine of mice harboring heterozygous mutations in Apc, a tumor suppressor gene [40]. Mechanistically, SETD2 knockdown changes the alternative splicing profile in cancer-associated genes and decreases intron retention in disheveled segment polarity protein 2 (DVL2), resulting in an increase in DVL2 protein, leading to activation of Wnt signaling. In non-small cell lung carcinoma, AKT isoforms (AKT1 and AKT3) phosphorylated Ser720/Thr721 in IWS1, an RNA processing regulator, promoting the recruitment of SETD2 to the RNA polymerase II complex [41].

IWS1 phosphorylation is correlated with the FGFR-2 splicing pattern, which is associated with epithelial–mesenchymal transition (EMT) and metastasis, in 21 of 24 non-small cell lung carcinoma specimens. These findings indicate that aberrant chromatin modification results in tumor malignancy and tumorigenesis through changes in the alternative splicing profile.

It has been reported that, in addition to epigenetic modifications that alter splicing patterns, aberrant splicing alters chromatin structure. For example, the mutation in SF3B1 leads the mutated SF3B1 to recognize the aberrant BPS within BRD9, a member of the chromatin remodeling complex called the noncanonical BAF complex, and promotes poison exon inclusion, resulting in the subsequent degradation of BRD9 mRNA (Figure 1). Deficiency in BRD9 affects specific loci in the genome, leading to abnormal gene expression and accelerated melanoma development [42].

## 5. Mutations in Splicing Factors Affect Cancer

Splicing abnormalities affect many genes, and aberrant alternative splicing alters the splicing patterns and gene expression in cancer. For example, comprehensive analysis of RNA sequencing showed that different splicing events occurred in breast cancer not only between cancer and normal tissues but also between different molecular types, such as triple-negative cells, non-triple-negative cells, and HER2-positive cells. Moreover, in breast cancer, alternative splicing signatures for epithelial–mesenchymal transition are neatly classified by basal and luminal subtypes [43]. Two thousand genes in CLL have also exhibited CLL-specific splicing isoforms between cancer and normal cells [44]. In addition, nearly 80 splicing events have been reported to be specifically observed in cancers with SF3B1 mutations, many of which involve the recognition of ectopic 3′ ss. Abnormal recognition of the 3′ ss has been confirmed in breast cancer and uveal melanoma with mutations in SF3B1 [45].

Mutations in the RNA splicing factor cause abnormalities in RNA splicing that affect cancer. Mutations in RNA splicing factors are more common in hematopoietic tumors such as myelodysplastic syndromes (MDS) and CLL, and are found in solid tumors such as uveal melanomas, pancreatic ductal carcinomas, and lung adenocarcinomas, but are less common than in hematopoietic tumors. SF3B1, SRSF2, U2AF1, and ZRSR2 are the four most commonly reported splicing factor mutations (Table 1). Mutations in SF3B1, SRSF2, and U2AF1 are concentrated as heterozygous mutations at the site coding for a particular amino acid, called a hot spot, and promote carcinogenesis or cancer progression by causing mis-sense mutations that alter the gain or alteration of the functional form of the protein. In contrast, ZRSR2 is a tumor suppressor gene and is considered to be a loss-of-function mutation, as it results in decreased protein expression that affects cancer. Mutations in these splicing factors are mutually exclusive because these simultaneous mutations probably induce cell death owing to synthetic lethality caused by functional alterations [46,47].

SF3B1, a subunit of U2 snRNP that binds to the BPS, is required for the spliceosome to recognize the 3′ ss. SF3B1 mutations were discovered via whole-exon sequencing in MDS with ring sideroblasts, which are pathologically highly characteristic. Mutations have been found in more than 70% of patients with MDS with this phenotype [48,49]. Mutations in SF3B1 have also been found in CLL, acute myeloid leukemia, uveal melanoma, and breast cancer. In clinical phenotypes of SF3B1 mutations, MDS with SF3B1 mutations have a relatively good prognosis, whereas CLL has a poor prognosis and is associated with chemotherapy resistance [50,51]. Mutation hotspots are found in the HEAT domain, but the normal function of this domain is not yet known. In cancer, mutated SF3B1 misrecognizes the polypyrimidine tract and binds to ectopic BPSs; thus, spliceosomes preferentially recognize upstream ectopic 3′ ss rather than normal 3′ ss [45,52,53] (Figure 1). The mechanism by which SF3B1 mutations alter the recognition of 3′ ss and the use of ectopic BPSs is not well understood. However, global splicing changes occur in tumors harboring SF3B1 mutations. Changes in the priority of 3′ ss associated with mutant SF3B1 resulted in splicing alterations that were clearly different from those inhibited by SF3B1 loss or pharmacological inhibition [53,54].

SRSF2 promotes exon skipping by binding to the ESE motif in exons. Mutations in SRSF2 are found in chronic myelomonocytic leukemia, acute myeloid leukemia, and high-risk MDS [55] and are concentrated in proline at position 95 (P95). SRSF2 generally bind to both C-rich ESE and G-rich ESE in a similar manner, but mutated SRSF2 preferentially recognizes C-rich CCNG ESE over G-rich [56,57,58], resulting in the enhancement of the recognition of exons with C-rich ESE and the suppression of the recognition of exons with G-rich ESE [56,58]. The change in ESE recognition causes aberrant splicing of hundreds of genes. For example, mutations in SRSF2 cause exon skipping in the mRNA EZH2 which regulates histone methylation, resulting in the creation of a new stop codon. These EZH2 mRNAs with exon skipping are subsequently degraded through nonsense-mediated decay, reducing EZH2 protein expression and H3K27me3 levels [58]. Loss of EZH2 function may be involved in MDS development and self-replication of abnormal hematopoietic stem cells [59,60,61].

Wild-type U2AF1 specifically recognizes and binds to the AG sequence of 3′ ss, acts in association with the U2 complex early in splicing, and is involved in the recruitment of the U2 complex [62,63]. U2AF1 mutations have been reported in MDS and lung cancer, and are primarily concentrated in two zinc finger domains, serine at position 34 (S34) and glutamine at position 157 (Q157), which are hotspots in the protein. Mutated U2AF1 recognizes a different site as 3′ ss [64] and targets a number of downstream genes, as shown in studies using patient transcriptomes, transgenic mouse models of U2AF1^S34F^, and human cells with U2AF1 mutations that affect residues S34 and Q157. The genes targeted for mis-splicing are classified as biological pathways for DNA damage, epigenetic regulation, and apoptosis [64,65].

Although it has been clarified that gene mutations in these splicing factors are involved in carcinogenesis by causing different and specific aberrant splicing, recent studies have demonstrated that there is a unifying mechanism underlying the etiology of MDS harboring mutations in SF3B1, SRSF2, and U2AF1. Mutations in splicing factors induce the accumulation of R-loops in MDS, formed from transcribed RNA-DNA hybrids and non-transcribed single-stranded DNA, leading to DNA damage, replication stress, and the activation of the ataxia telangiectasia Rad3-related protein (ATR)-Chk1 pathway. Mutations in these splicing factors have also been suggested to have therapeutic vulnerabilities, such as greater sensitivity to specific molecular inhibitors, and are expected to be a novel therapeutic strategy [66,67,68].

ZRSR2 is essential for 3′ ss recognition of U12-type splicing of the minor spliceosome [69]. ZRSR2 mutations are frequently found in MDS, a type of acute myeloid leukemia, a part of T-cell ALL, and thyroid cancer. Unlike SF3B1, SRSF2, and U2AF1, ZRSR2 mutations are loss-of-function mutations. The ZRSR2 mutation induces mis-splicing involved in U12-type intron retention, resulting in nonsense or frameshift mutations. Thus, this mutation decreases the production of ZRSR2, affecting the MAPK pathway and E2F transcription factors [70]. However, further studies are required to elucidate how mis-splicing owing to deletion of ZRSR2 leads to tumorigenesis.

## 6. Genomic Mutations Create New Splice Sites and Alter Splicing Patterns

Genomic mutations in splicing factors and trans-acting factors cause aberrant splicing that affects the development and progression of cancer, whereas mutations in cis-regulatory factor sequences alter splicing patterns, such as exon skipping and intron retention, through the recognition of ectopic splice sites by spliceosomes on pre-mRNA (Table 2).

RNA-seq and exome data identified single-nucleotide variants (SNVs) that induce aberrant splicing, such as exon skipping and intron retention. SNVs that cause intron retention occur in tumor suppressor genes For example, SNVs in *TP53* were found in breast carcinoma, colorectal carcinoma, lung squamous cell carcinoma, and ovarian serous cystadenocarcinoma [71]. Most of these SNVs create a premature termination codon, which reduces the tumor suppressive function by loss of function through nonsense-mediated decay and protein truncation. RNA seq analysis of lung adenocarcinoma by TCGA also showed that splice site mutation and deletion in the oncogene *MET* resulted in exon14 skipping [72], which causes stabilization of the protein, followed by MET activation.

In addition, bioinformatics analyses using whole-genome sequencing and whole-exome sequencing have recently revealed that somatic mutations in non-coding sequences cause splicing changes and generate new exons. The introns associated with the creation of new exons were predominantly larger than the genome-wide mean intron length, and splicing changes induced by non-coding mutations were observed in cancer-related genes such as *ATRX, BCOR, CDKN2B, MAP3K1, MAP3K4, MDM2, SMAD4, STK11*, and *TP53*, leading to truncated protein production and gene expression alteration [73]. These new exons are thought to be created through alternative splicing because a non-coding mutation switches the original potential splice site to a functional splice site. As bioinformatics continues to develop, it has become increasingly clear that mutations in genetic sequences as cis-regulatory factors alter splicing patterns and affect cancers.

## 7. Differential Expression of Splicing Factors Affect Cancer

Changes in the expression of splicing factors, such as SR proteins and hnRNPs, have also been reported to affect cancer (Table 1). Some SR proteins are overexpressed in cells and act as oncoproteins; for example, SRSF1 is upregulated in lung, colon, and breast cancers [74,75] and its overexpression induces immortalization of mouse fibroblasts and human and mouse mammary epithelial cells [74,75]. Overexpression of SRSF1 induces mis-splicing of MNK2 and S6K1 and promotes transformation by activating the mTOR pathway, which is essential for SRSF1-mediated transformation. In addition, overexpression of SRSF1 promotes alternative splicing of BIM and BIN1; the produced isoforms thus lose their pro-apoptotic functions. In addition, SRSF3 regulates the alternative splicing of TP53; its decreased expression induces the production of p53β, an isoform of p53, and promotes p53-mediated cellular senescence [76]. These results indicate that SRSF1 acts as an oncoprotein that exacerbates cancer, and that SRSF3 promotes carcinogenesis, cancer progression, and maintenance of growth through its overexpression [77].

The expression of hnRNPs may act as a tumor-promoting or anti-tumor agent in cancer. MYC, a transcription factor, is upregulated in various cancers and is involved in the expression of many splicing factors and aberrant splicing. The upregulation of some specific hnRNPs, hnRNP A1, hnRNP A2, and PTB, is involved in the splicing regulation of pyruvate kinase (PKM) and promotes the expression of cancer-associated PKM2 isoforms [78,79]. There are two mutually exclusive PKM isoforms: PKM2 is upregulated in cancer, promoting aerobic glycolysis, whereas PKM1 is expressed in most normal tissues, promoting oxidative phosphorylation. hnRNP A1 regulates alternative splicing of MYC-associated factor X (MAX) to produce delta MAX. Upregulated hnRNP A1 enhances delta MAX expression and promotes MYC-dependent transformation, glycolytic gene expression, and tumor growth [80,81]. MYC also regulates the expression of hnRNP H, which controls ARAF kinase splicing [82] and increases the expression of isoforms that promote RAS-induced transformation. In contrast, regarding hnRNP K, myelopoietic transformation was shown in a mouse model harboring an Hnrnpk knockout allele and decreased hnRNP K expression reduced p21 activation and C/EBP expression levels, which activated STAT3 signaling, suggesting that it acts as a tumor suppressor [83]. However, the mechanism of this transformation is still unclear, as hnRNP K is involved in many biological processes, including RNA splicing.

## 8. Splicing Variants Induced by Aberrant Splicing

Prostate cancer is one of the most common causes of cancer-related deaths in men. Although most patients with localized prostate cancer show a good response to treatment, progressive and metastatic prostate cancer results in approximately 307,000 deaths in patients. Androgen deprivation therapy is the first-line therapy for patients with metastatic prostate cancer. Therapy targets the ligand-binding domain at the C-terminal domain of the androgen receptor (AR) to inhibit AR-dependent gene expression. This therapy is effective for patients at the onset; however, most patients eventually develop castration-resistant prostate cancer (CRPC).

Prostate cancer expresses various truncated AR variants that lack the C-terminal domain through aberrant splicing regulation. AR-V7 has attracted considerable attention because of its association with aggressive prostate cancer and CRPC progression. AR-V7 shares the N-terminal domain, including the transcription-activating domain and DNA-binding domain with the full-length AR (AR-FL) and has, instead of its normal terminus, 16 AR-V7-specific amino acids that are encoded on the cryptic exon3 in AR. AR-V7 is considered to be a constitutively active form, resulting in androgen-independent prostate cancer cell proliferation (Table 1). AR-V7, however, has been shown to inhibit transcription and support CRPC growth by repressing growth-suppressive genes in conjunction with AR-FL [84]. AR-V7 expression in circulating prostate cancer cells is also correlated with resistance to enzalutamide and abiraterone, inhibitors of AR signaling [85], indicating a critical role of AR-V7 in CRPC progression.

The molecular mechanisms of AR-V7 splicing have recently been elucidated by several groups, including ours. AR-V7 expression is reported to be correlated with several RNA splicing factors, such as U2 small nuclear RNA auxiliary factor 2 (U2AF2), KH domain-containing RNA-binding, signal transduction associated 1 (KHDRBS1, also known as Sam68), splicing factor proline and glutamine rich (PSF, also known as SFPQ), non-POU domain-containing octamer-binding (NONO), and cleavage and polyadenylation specific factor 1 (CPSF1) [86,87,88,89]. AR-V7 splicing may be more vulnerable than general splicing, which may be one of the reasons A-V7 expression is affected by the knockdown of several RNA splicing factors.

We have shown direct evidence that splicing factor 3b subunit 2 (SF3B2, also known as SAP145 and SF3b145) directly binds to the cryptic exon3 in the *AR* gene at nucleotide resolution using the PAR-CLIP and CRISPR/Cas9 system (Figure 1). Furthermore, pladienolide B disrupts the SF3B2 complex and represses tumor growth during castration. The structure of SF3B2 has not been determined because of its highly disordered domains [90]. Recently, the molecular architecture of the 17S U2 snRNP containing the SF3b complex was identified, revealing the position of SF3B2 in the complex [91]. The disordered domain consisting of amino acids 531–564 may be structurally regulated by SF3B1 and TAT-SF1 in the U2 snRNP complex, which recognizes the BPS in the 3′ region of the intron; SF3B2 nonetheless binds to some specific exons but not all introns, unlike U2AF2, as it is detected at most of the 3′ splice site [92]. Taken together, SF3B2 may be able to bind to the target RNA sequence when its disordered domain is structurally opened by the movement of other components in the complex. Indeed, in comparison to the SF3B1 complex, the SF3B2 complex lacks some components, suggesting that the structure is critical for the interaction between SF3B2 and RNA.

Many studies have reported that alternative splicing and aberrant splicing are related to splicing variants related to signaling pathways, leading to cell proliferation and cell death in cancer cells (Table 2). Splicing variants of RAF downstream of KRAS have been reported. BRAF has two variable exons, 8b and 10, which can generate four different isoforms [93]. Variants containing exon 10 activate downstream MEK1/2, whereas variants containing exon 8b exhibit the opposite response. In addition to activating BRAF mutations, cancer-associated BRAF splicing variants caused by aberrant splicing have been reported. Thyroid carcinomas express splicing variants of BRAF that lack the N-terminal auto-inhibitory domain, which results in constitutive BRAF activation and in turn activates the MAP kinase signaling pathway [94]. In melanomas with the BRAF^V600E^ mutation, a variant lacking exons 4–8 was identified that produces a BRAF^V600E^ protein lacking the RAS binding domain, which enhances dimerization and confers resistance to the ATP-competitive BRAF inhibitor vemurafenib [95].

PTEN, a tumor suppressor gene that represses PI3K activity, is generated in several isoforms by alternative splicing. PTEN5b, which retains intron 5b, is one of these isoforms and is upregulated in breast cancer. This PTEN5b acts as a dominant-negative and consequently enhances PI3K activation, contrary to the effect of PTEN [96]. mTORβ, the splicing isoform of mTOR downstream of AKT, is the activating form that regulates the G1 phase of the cell cycle and promotes cell proliferation, in contrast to the full-length protein (mTORα) [97]. Ribosomal S6 kinase 1 (S6K1) is a signaling molecule downstream of mTOR that regulates cell size and translational efficiency. S6K1 undergoes alternative splicing to produce long and short isoforms. SRSF1 promotes the production of the short isoforms, S6K1 h6A and h6C, which are upregulated in breast cancer and induce the transformation of human mammary epithelial cells. These short isoforms activate mTORC1, which acts as an oncogenic isoform. In contrast, long isoforms have the opposite effect of inhibiting RAS-induced transformation and tumorigenesis. These findings suggest that alternative splicing of S6K1 may act as a molecular switch at the crossroads of tumor enhancement or antitumor activity in breast cancers [98].

EGFR, an upstream receptor tyrosine kinase that activates MAPK signaling, also undergoes alternative splicing to produce splicing variants. One EGFR variant, EGFRvIII, lacks exons2–7 and part of the domain where extracellular ligands bind and is constitutively activated to promote cell proliferation [99,100]. The isoform of EGFR that lacks exon4 (de4 EGFR) is also expressed in several cancers and is constitutively activated to enhance transformation and promote metastasis [101]. Hepatocyte growth factor activates proliferative signaling by binding to the receptor tyrosine kinase MET. In cancer, these genes are amplified or mutated to activate MET and promote tumorigenicity. MET also undergoes alternative splicing to produce a variant with skipping exon14 (METex14). Exon14 of MET contains Y1003 in a DpTR motif, which is a target for ubiquitin degradation. Loss of this motif increases MET stability, prolongs hepatocyte growth factor stimulation, and enhances oncogenic activation. METex14 expression is detected in lung cancers and gliomas, and its expression enhances sensitivity to MET inhibitors [102].

## 9. Treatment Targeting Splicing

Compounds that inhibit splicing catalysis or post-translational modifications of splicing factors have been developed for various cancers. Mutations in splicing factors, such as SF3B1, SRSF2, and U2AF1, are common in hematopoietic tumors, and cancer cells with these heterozygous mutations survive depending on the function of wild-type splicing factors, which are expressed in a mutually exclusive manner to prevent synthetic lethality [46,47]. In addition, hematopoietic cells with SF3B1, SRSF2, and U2AF1 mutations are more sensitive to SF3B inhibitors [46,103,104]. Based on these results, compounds that inhibit splicing factors have been developed.

Spliceostatin A and pladienolide B have been reported as natural compounds that change splicing by combining with SF3b. E7107 and H3B-8800 were identified as small molecules that competitively bind to the SF3b complex with these natural compounds. Drugs such as pladienolide B and E7107 inhibit splicing by binding to the pocket of the SF3b complex that binds to the BPS [105,106]. H3B-8800 preferentially caused cell death in SF3B1-mutant epithelial and hematologic tumor cells [54], showing that H3B-8800, but not E7107, is highly selective for cells harboring a mutant SF3B1 gene. Furthermore, cell lines that acquired resistance to these agents were established by long-term culture in the presence of pladienolide B or H3B-8800. The exome sequence analysis of these cell lines identified the Arg1074His mutation in SF3B1 or Trp36Cys mutation in PHF5A [107]. As PHF5A is known as a component protein of the SF3b complex that is involved in the recognition of BPS, these results also show the high selectivity of H3B-8800 to the SF3b complex.

The clinical trial for E7017 was discontinued because of the side effects of optic nerve damage [108,109]. We developed two drugs (phenyl-C-glycoside analogue and 1,2-deoxy-pyranose analogue) that improved spliceostatin A, showing that they suppressed the expression of AR-V7 and cell proliferation and could also avoid severe toxicity compared to spliceostatin A in vivo [110,111]. In addition, a clinical trial for H3B-8800 is currently underway (NCT02841540) (Table 3). Clinical trials of GSK3326595, an inhibitor of protein arginine methyltransferase 5 (PRMT5) are currently underway (NCT04676516, NCT03573310, NCT03854227, NCT02783300, NCT03614728) (Table 3). PRMT5 has type II PRMT activity to symmetric dimethylate arginine residues in the target proteins, including histones, and a splicing regulating factor SmD3 [112,113,114,115,116,117,118]. PRMT5 overexpression is associated with tumorigenesis and poor prognosis. Mechanistically, PRMT5 is involved in repressing the expression of tumor suppressor genes. As PRMT5 knockdown results in a decrease in cell proliferation in cancer cell lines [112,113,119,120,121], PRMT5 is thought to be a good target protein for developing anti-cancer drugs against PRMT5-overexpressed tumors. GSK3326595 (EPZ015666) was identified by high-throughput screening of compounds that inhibit PRMT5 enzymatic activity [122]. GSK3326595 shows anti-tumor effects in vivo and in vitro, along with decreased methylation of SmD3 but not histones. GSK3326595 alters alternative splicing of MDM4, resulting in activation of the p53 pathway [123]. These findings suggest that GSK3326595 provides clinical benefits to patients with lymphoma and solid tumors.

Sulfonamide compounds such as E7070 (indisulam) and E7820, which have been developed as anticancer agents for many years, have recently been reported as new therapeutic agents to control splicing, thus revealing some of their molecular targets and functions. The sulfonamide compound selectively degrades splicing factor RBM39 (CAPERα) by combining with DCAF15, the substrate recognition protein of the CUL4 ubiquitin ligase complex. RBM39 is an RNA-binding protein associated with the U2AF complex, and its degradation induces intron retention and exon skipping [124,125]. The mechanisms of ubiquitin ligase-mediated splicing regulation in these compounds may suggest promising new therapeutic strategies in the future.

Splicing regulation by nucleic acids is a therapeutic method to control splicing in a sequence-specific manner. Nucleic acids comprise oligo-nucleic acids linked to more than ten to several tens of bases and act directly on living organisms without involving gene expression. Currently, antisense, siRNA, aptamers, and CpG oligos are in practical use. Two major mechanisms of action are known for nucleic acids, complementary binding to RNA or proteins, resulting in the inhibition of their function as therapeutic agents. Similar to nucleic acids that control splicing, antisense oligonucleotides that inhibit the binding of splicing factors to pre-mRNA have been developed, e.g., oligonucleotides for Duchenne muscular dystrophy and spinal muscular atrophy [126,127,128,129]. Duchenne muscular dystrophy is a disease in which a specific exon of the gene for dystrophin, which is essential for the maintenance of muscle cells, is deleted. Deletion of this exon induces mRNA truncation and dysfunctional protein synthesis, resulting in impaired muscle function. Exon skipping is induced by introducing a complementary antisense to other exons, which are not directly related to the function; functional proteins are synthesized by avoiding truncation and muscle function is recovered [130].

Basic research has shown that nucleic acids can exert anti-tumor effects in cancers. For example, signal transducer and activator of transcription 3 (STAT3) is a transcription factor that activates multiple oncogenic pathways and is active in various cancers, including breast cancer. STAT3α, a splicing variant of STAT3, has oncogenic properties, whereas the splicing variant STAT3β undergoes alternative splicing to be a truncated isoform lacking the C-terminal trans-activation domain and acts as a dominant negative transcriptional regulator to induce apoptosis. Antisense nucleotides alter alternative splicing, resulting in an expression shift of endogenous STAT3α to STAT3β. The induction of STAT3β was shown to lead to apoptosis and cell cycle arrest when compared to full STAT3 knockdown in cell lines with persistent STAT3 tyrosine phosphorylation; it also suppressed tumor growth in vivo [131]. However, it has not yet been utilized for the practical applications of nucleic acid drugs. The lack of progress in the development of nucleic acid drugs is considered to be the inefficient delivery of drugs to the tumor tissue. Recently, the first systemically administered nucleic acid for familial hypercholesterolemia was launched. Since then, systemic nucleic acid drugs administered via transvenous or oral routes have been developed. This rapid progress in delivery technology has led to the practical application of nucleic acids, which are expected to become a next-generation drug. Nucleic acids are expected to have high specificity and efficacy similar to therapeutic antibodies, with the advantage that they can also be produced by the chemical synthesis of small molecule compounds. In addition, because their base drug substance is easily adaptable oligo nucleic acids, it is expected that the possible drug targets will quickly expand after refinement of the technology and its first successful application.

Many recent studies have revealed that aberrant regulation of splicing is involved in tumorigenesis and progression of malignancy. However, the molecular mechanisms by which cancer cells change their splicing profile to promote tumor growth and resistance to treatment remain unclear. Although various mutations in genes related to splicing have been detected in hematologic cancer cells, these mutations are rarely found in solid tumors. Several splicing factors, such as SF3B2, are involved in the generation of splicing variants. The high splicing factor expression results in changes in the splicing profile; nonetheless, the mechanisms that increase the expression of splicing factors are unknown in most cases. It is still unclear whether the expression levels of splicing factors change splicing factor-binding RNA regions, splicing factor complexes, and transcription. The splicing switch mechanisms that drive malignancy remain unclear. Elucidating these mechanisms may allow us to identify better therapeutic targets for cancer.

## Figures and Tables

**Figure 1 cells-10-00923-f001:**
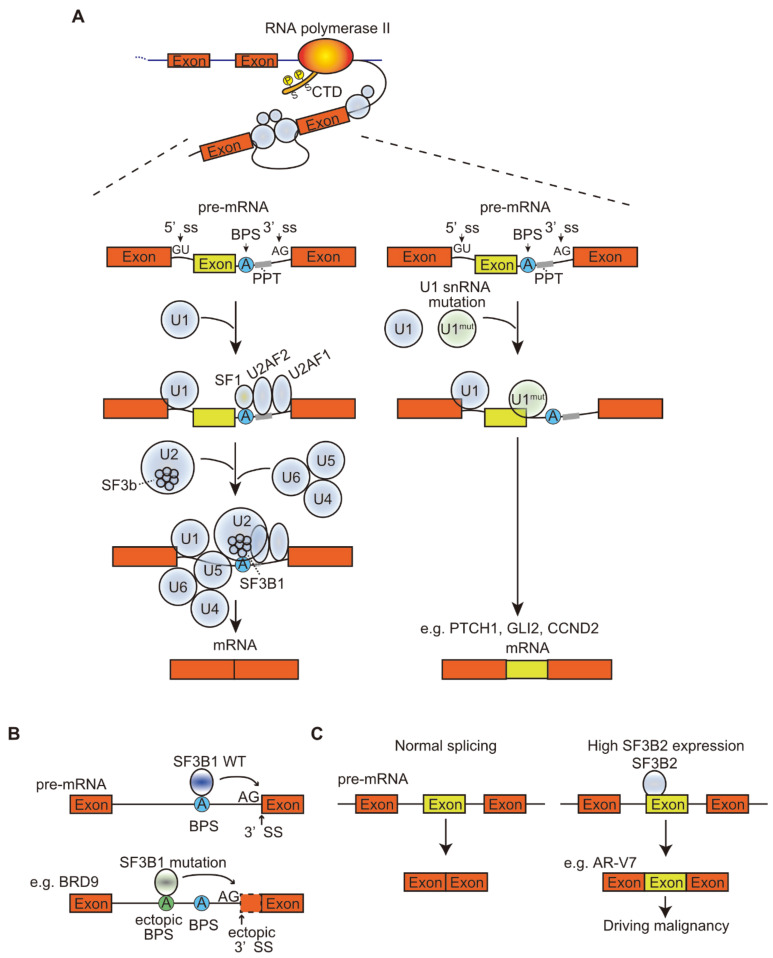
Splicing is dysregulated by mutations in snRNA and splicing factors and expression levels of splicing factors. Splicing is a catalytical process that removes introns from precursor mRNA (pre-mRNA) to generate mature mRNA and is caused by dynamic changes in large protein-RNA complexes called spliceosomes. pre-mRNA is processed by the spliceosome in conjunction with transcription regulating factors. (**A**) The pre-mRNA is transcribed by RNA polymerase II (RNAPII) and spliced by spliceosomes. In the major spliceosomes, U1 and U2 small nuclear ribonucleoproteins (snRNPs) have a pivotal role to recognize 5′ and 3′ splice sites (ss). These complexes recognize functional intron sequences such as 5′ and 3′ ss, the branch point site (BPS), and the polypyrimidine tract (PPT). The U1 snRNP binds to the 5′ ss on the pre-mRNA and the U2 snRNP binds to the BPS with the aid of the U2 auxiliary factor (U2AF). This splicing machinery in association with other proteins removes the intron with a lariat structure from pre-mRNA to generate the mature mRNA. The mutation in U1 snRNA modifies the RNA recognition sequence of U1 snRNP, allowing the mutated U1 snRNP to bind to the 5′ ss of cryptic exon, which results in the inclusion of the cryptic exon. (**B**) SF3B1 is a critical factor to bind the SF3b complex to the BPS sequence in the intron. A mutation in SF3B1 modifies the RNA recognition sequence, allowing mutated SF3B1-binding to an ectopic BPS. The mis-binding shifts the position of the 3′ ss upstream, resulting in ectopic 3′ ss. (**C**) Expression levels of splicing factors are also involved in the regulation of splicing. SF3B2 is a component of the SF3b complex that is ubiquitously expressed. High SF3B2 expression promotes splicing in the target exons and introns, resulting in increased cancer malignancy-driving splicing variants, such as AR-V7.

**Table 1 cells-10-00923-t001:** Aberrant splicing depending on mutations or different expression of splicing factors in cancer.

Splicing Factor	Aberrant Splicing Alter the Mechanism in Cancer
**snRNA mutation associate with aberrant alternative splicing**	
U1 snRNA	A>C mutation in CLL and HCC and A>G in SHH medulloblastomas at the third base of U1 snRNA. The mutated U1 snRNA recognize the ectopic 5′ss, resulting in the alteration the splicing patterns.
**Mutations in splicing factors affect cancer**	
SF3B1	Mis-recognizes polypyrimidine tract (PPT) and bind to ectopic the branch point site (BPS) in MDS, CLL, AML, uveal melanoma and breast cancer
SRSF2	Preferentially recognize C-rich ESE over G-rich ESE, leading to aberrant splicing in CMML, AML and high-risk MDS
U2AF1	Recognizes different site as 3’ ss in MDS and lung cancer
ZRSR2	Induces U12-type mis-splicing leading to intron retention in MDS, a type of AML, a part of T-cell ALL and thyroid cancer. ZRSR2 mutation indicates the loss-of-function mutation.
**Differential expression of splicing factors affect cancer**	
SF3B2	Promotes AR-V7, an isoform of AR, that lacks the C-terminal domain. AR-V7 is a constitutively active form, resulting in androgen-independent prostate cancer cell proliferation.
SRSF1	Promotes isoforms of MNK2 and S6K1, activating the mTOR pathway. SRSF1 overexpression also promotes alternative splicing of BIM and BIN1: the produced isoforms lose their pro-apoptotic functions.
SRSF3	Regulates the alternative splicing of TP53: SRSF3 loss induces the production of p53β, an isoform of p53, and promotes p53-mediated cellular senescence.
hnRNP A1	Promotes PKM2, an isoform of PKM, leading to aerobic glycolysis in cancer, and delta MAX, an isoform of MAX, promoting glycolytic gene expression and tumor growth
hnRNP H	Controls ARAF kinase splicing and increases the expression of isoforms that promote RAS-induced transformation
hnRNP K	Acts as a tumor suppressor in leukemia: decreased hnRNP K expression resulted in decreased p21 activation and C/EBP expression levels, which activated STAT3 signaling
RBM4	Acts as a tumor suppressor in various cancer cells by promoting the pro-apoptotic isoform BCL-X_S_ of BCL2L1 and antagonizing the oncogenic effects of SRSF1 on mTOR activation
RBM5	Modulates apoptosis by regulating alternative splicing of CASP2 and FAS

**Table 2 cells-10-00923-t002:** Aberrant splicing depending on gene mutations or splicing variants in cancer.

Gene	Aberrant Splicing Alter the Mechanism in Cancer
**Genomic mutations create new splice sites and alter splicing pattern**	
*TP53*	Mutation of exons in *TP53* creates a premature termination codon, which reduces the tumor suppressive function in breast cancer, colorectal cancer, lung squamous cell carcinoma, and ovarian serous cystadeno carcinoma.
*MET*	Mutation and deletion of splice site in *MET* induces Exon14 skipping in lung adenocarcinoma
Non-coding	Produces truncated proteins, which alter gene expression in some gens such as *ATRX, BCOR, CDKN2B, MAP3K1, MAP3K4, MDM2, SMAD4, STK11*, and *TP53*.
**Aberrant splicing associate with signaling pathways**	
BRAF	Variant that contains exon10 but not 8b leads to the MEK1/2 activation.
	Variant that contains exon8b but not 10 leads to the MEK1/2 suppression.
	Variant that lacks the N-terminal auto-inhibitory domain leads to constitutive BRAF activation and the following activation of the MAP kinase signaling pathway in thyroid carcinoma.
	Variant that lacks exons4-8 in melanomas with the BRAF^V600E^ mutation leads to the inhibition of RAS and the resistance to the ATP-competitive BRAF inhibitor vemurafenib.
PTEN	PTEN5b that retains intron5b in breast cancer acts as a dominant-negative and leads to consequently PI3K activation.
mTOR	mTORβ, a short isoform of mTOR, promotes the G1 phase of the cell cycle and cell proliferation.
Ribosomal S6 kinase1 (S6K1)	S6K1 h6A and h6C, short isoforms of S6K1 in breast cancer, induce the transformation in human mammary epithelial cells and activate of mTORC1, which acts as an oncogenic isoform.
EGFR	EGFRvIII, lacking exons2–7 of EGFR, promotes constitutive activation to promote cell proliferation.
	de4 EGFR, lacking exon4 of EGFR, promotes constitutive activation to enhance transformation and metastasis.
MET	METex14 that lacks exon14 of MET in lung cancer and glioma increases MET stability, prolongs HGF stimulation, and oncogenic activation because exon14 is a target for ubiquitin degradation.

**Table 3 cells-10-00923-t003:** Ongoing clinical trials associated with targeting splicing processes.

No	Ongoing Clinical Trials (accessed date 1 March 2021)	
**1**	**Title**	A Phase II Window of Opportunity Trial of PRMT5 Inhibitor, GSK3326595, in Early Stage Breast Cancer
	**Conditions**	Breast Cancer
	**Interventions**	Drug: GSK3326595
	**URL**	https://ClinicalTrials.gov/show/NCT04676516
**2**	**Title**	A Study of JNJ-64619178, an Inhibitor of PRMT5 in Participants With Advanced Solid Tumors, NHL, and Lower Risk MDS
	**Conditions**	Neoplasms, Solid Tumor, Adult, Non-Hodgkin Lymphoma, MDS
	**Interventions**	Drug: JNJ-64619178
	**URL**	https://ClinicalTrials.gov/show/NCT03573310
**3**	**Title**	A Dose Escalation Study Of PF-06939999 In Participants With Advanced Or Metastatic Solid Tumors
	**Conditions**	Advanced Solid Tumors, Metastatic Solid Tumors
	**Interventions**	Drug: PF-06939999 dose escalationDrug: PF-06939999 monotherapyDrug: PF-06939999 in combination with docetaxel
	**URL**	https://ClinicalTrials.gov/show/NCT03854227
**4**	**Title**	Dose Escalation Study of GSK3326595 in Participants With Solid Tumors and Non-Hodgkin’s Lymphoma (NHL)
	**Conditions**	Neoplasms
	**Interventions**	Drug: GSK3326595Drug: Pembrolizumab
	**URL**	https://ClinicalTrials.gov/show/NCT02783300
**5**	**Title**	Study to Investigate the Safety and Clinical Activity of GSK3326595 and Other Agents to Treat Myelodysplastic Syndrome (MDS) and Acute Myeloid Leukemia (AML)
	**Conditions**	Neoplasms
	**Interventions**	Drug: GSK3326595Drug: 5-azacytidineDrug: Best available care
	**URL**	https://ClinicalTrials.gov/show/NCT03614728
**6**	**Title**	A Phase 1 Study to Evaluate H3B-8800 in Participants With Myelodysplastic Syndromes, Acute Myeloid Leukemia, and Chronic Myelomonocytic Leukemia
	**Conditions**	MDS, AML, Chronic Myelomonocytic Leukemia (CML)
	**Interventions**	Drug: H3B-8800
	**URL**	https://ClinicalTrials.gov/show/NCT02841540

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
