# Peer review of "Regulation of RNA Splicing: Aberrant Splicing Regulation and Therapeutic Targets in Cancer"

_cells, 2021, doi:10.3390/cells10040923_

Round 1
Reviewer 1 Report
The quality of the figure and tables must be improved.
Generalities about splicing are too deep and don't seem necessary to understand the impact of splicing on cancer.
Section 4 is too short.
Sections 5 to 8 are presented in a tedious way. This could be improved by summarizing the information either in tables or figures.
Additional and more recent references must be included.
Author Response
The quality of the figure and tables must be improved.
Reply: We modified the figure and tables. (page 2-3, lines 54-68)
Generalities about splicing are too deep and don't seem necessary to understand the impact of splicing on cancer.
Reply: We shortened the explanation of generalities about splicing.
Section 4 is too short.
Reply: We merged section 4 with section 5.
Sections 5 to 8 are presented in a tedious way. This could be improved by summarizing the information either in tables or figures.
Reply: We modified tables to help understanding for sections 5 to 8.
Additional and more recent references must be included.
Reply: We added more recent references including works regarding snRNA. (page 3-4, lines 90-146)
Reviewer 2 Report
Globally the manuscript is well written and organized. However it looks like an old-school and basic review of splicing alterations in cancer diseases. This is also typified by relevant but quite old references of the litterature. What I am expecting from a new published review is a an updated state-of-the-art of the field, which is unfortunately not the case here.
The authors ignore new concepts in the field like mutations in snRNA (e. U1 snRNA), differential expression of snRNAs in cancers and modulated expression of guide sca/snoRNA linked to snRNA activation in several types of cancers, aberrant splicing events such intron retention,…
I strongly invite the authors to further develop these emerging concepts. The review will be then more exciting for the splicing and spliceosome community and will significantly increase its citation impact.
To make more space in the manuscript, the section #3 related to splicing, transcription and chromatin could be reduced (or entirely removed). I also suggest to connect information in section#4 with section #5. In contrast key information is missing for a large uninformed audience. For instance, snRNP biogenesis and the post-transcriptionnal of snRNA (so important for spliceosome activation and splicing regulation !) must be mentionned somewhere.
The illustrations are ok but not terrific. A global overview of cancer–associated aberrations in splicing would be a plus value. I also invite the authors to present a table with the ongoing clinical trials targeting splicing factors.
A conclusion and a perspective section with open questions will help new readers to get a overview of the future research in the field.
I strongly suggest the authors to carrefully check several confusing and unclear sentences in the manuscript
For example :
- « splicing interacts temporally and spatially with transcriptional mechanisms » => I would say splicing is linked
- « SF3B1 mutations reduce BRD9 » => It is a too severe shorcut
- EZH2, which regulates histone methylation, induces aberrant splicing of pre-mRNA, leading to nonsense-mediated decay and the reduction of both EZH2 protein expression and H3K27me3 levels => This is totally unclear
- ….and many more
Finally I also noticed different types of police in the text, probably resulting from copy/paste text. Please revise and modify.
Author Response
Comments and Suggestions for Authors
Globally the manuscript is well written and organized. However it looks like an old-school and basic review of splicing alterations in cancer diseases. This is also typified by relevant but quite old references of the litterature. What I am expecting from a new published review is a an updated state-of-the-art of the field, which is unfortunately not the case here.
The authors ignore new concepts in the field like mutations in snRNA (e. U1 snRNA), differential expression of snRNAs in cancers and modulated expression of guide sca/snoRNA linked to snRNA activation in several types of cancers, aberrant splicing events such intron retention,…
I strongly invite the authors to further develop these emerging concepts. The review will be then more exciting for the splicing and spliceosome community and will significantly increase its citation impact.
Reply: Thank you for your kind comment. We added the topics about snRNA, such as U1 snRNA. (page 3-4, lines 90-146)
To make more space in the manuscript, the section #3 related to splicing, transcription and chromatin could be reduced (or entirely removed).
Reply: We shortened the section 3 as your comment.
I also suggest to connect information in section#4 with section #5.
Reply: Thank you for your kind comment. We combined section 4 with section 5.
In contrast key information is missing for a large uninformed audience. For instance, snRNP biogenesis and the post-transcriptionnal of snRNA (so important for spliceosome activation and splicing regulation !) must be mentionned somewhere.
Reply: We added the topics about snRNP biogenesis and post-transcriptional modification of snRNA. (page 3-4, lines 90-146)
The illustrations are ok but not terrific. A global overview of cancer–associated aberrations in splicing would be a plus value. I also invite the authors to present a table with the ongoing clinical trials targeting splicing factors.
Reply: We modified the figure as more cancer-associated aberrations in splicing. We also added the table showing the ongoing clinical trials targeting splicing. (page 2-3, lines 54-68), (page 13, lines 521-535)
A conclusion and a perspective section with open questions will help new readers to get a overview of the future research in the field.
Reply: We added a conclusion paragraph with open questions at the end of the manuscript. (page 14, lines 582-593)
I strongly suggest the authors to carrefully check several confusing and unclear sentences in the manuscript
For example :
- « splicing interacts temporally and spatially with transcriptional mechanisms » => I would say splicing is linked
- « SF3B1 mutations reduce BRD9 » => It is a too severe shorcut
- EZH2, which regulates histone methylation, induces aberrant splicing of pre-mRNA, leading to nonsense-mediated decay and the reduction of both EZH2 protein expression and H3K27me3 levels => This is totally unclear
- ….and many more
Finally I also noticed different types of police in the text, probably resulting from copy/paste text. Please revise and modify.
Reply: Thank you for your kind comments. We improved these sentences to easily understand.
- page 4-5, lines 150-151
- page 5, lines 173-176
- page 6, lines 222-227
- page 7, lines 301-305
- page 8, lines 329-332
- page 12, lines 508-515
Round 2
Reviewer 2 Report
The title of this review suggests that cancer-specific mechanisms in splicing regulation will be discussed, which is unfortunately missing or not well described. For instance it is unclear if the mechanisms described in sections #3 and #4 are common in splicing regulation or specific for cancer cells. I also strongly suggest the authors to revise the structure/organization of paragraphs and sections, and to get the manuscript edited for the language from a native English speaker.
In its present form, the paragraph on snRNP biogenesesis is unclear and not complete. There is no information about the mechanisms regulating snRNA modifications (guide sca/snoRNAs, snoRNP, …). A small but precise paragraph would be highly appreciated.
Pseudouridylation is also an important post-transcriptional modification of snRNA but it is unfortunately not mentionned in the text. The authors should better distinguish 2’-O-methylation and pseudouridylation on snRNAs versus different types of methylation on mRNAs. Recent literature reports that the modulated expression of guide sca/snoRNA can be linked to snRNA activation in several types of cancers. This must be discussed as previously suggested.
As mentioned above we are wondering if the described mechanism of action for FTO, METTL4/16 and CSTF2tau is specific and/or different in cancer cells?
I am not sure that the section#3 (now section#4) about splicing and chromatin has been reduced as requested. I am still not convinced that this section is mandatory and again, I don’t see a clear connection with cancer in this chapter. What are the specific epigenetic mechanisms involved in splicing regulation in cancer?
Section #7 : There are other spliced variants in other types of cancers (Met exon 14, BCL2, FAS ..). I propose to add a table with relevant examples. Please move the ARV7 paragraph in section #8. It would be more obvious.
Minor comments:
- U2 and U12 introns should be mentioned earlier
- BRD9 itself is not the ncBAF complex. It is one specific member.
- « are thought to be due to synthetic lethality caused by their functional alterations » Please can you clarify this point ?
- characteristic: ? « : » is probably a typo Line 281
« are concentrated in P95 ». What is P ? I assume it is a proline residue at position 95 ? Same comments for S34 and Q157 ? Please change concentrate
« the isoforms thus produced lose their pro-apoptotic functions ». Please change the text. Line 428
- Figure 1: This is a too schematic view of spliceosome activation and splicing regulation ; It does not only depends on U1. Please add the other U snRNAs in the figure
Author Response
The title of this review suggests that cancer-specific mechanisms in splicing regulation will be discussed, which is unfortunately missing or not well described. For instance it is unclear if the mechanisms described in sections #3 and #4 are common in splicing regulation or specific for cancer cells. I also strongly suggest the authors to revise the structure/organization of paragraphs and sections, and to get the manuscript edited for the language from a native English speaker.
Reply: We revised sections #3 and #4 to focus on cancer-related topics. We also reduced the volume of section #4. This manuscript was proofread in English.
In its present form, the paragraph on snRNP biogenesesis is unclear and not complete. There is no information about the mechanisms regulating snRNA modifications (guide sca/snoRNAs, snoRNP, ...). A small but precise paragraph would be highly appreciated.
Pseudouridylation is also an important post-transcriptional modification of snRNA but it is unfortunately not mentionned in the text. The authors should better distinguish 2’-O- methylation and pseudouridylation on snRNAs versus different types of methylation on mRNAs. Recent literature reports that the modulated expression of guide sca/snoRNA can be linked to snRNA activation in several types of cancers. This must be discussed as previously suggested.
Reply: We added the description of snRNA modification including 2'-O-methylation and pseudouridine through guide sno/scaRNA. (page.3, lines.105-133)
As mentioned above we are wondering if the described mechanism of action for FTO, METTL4/16 and CSTF2tau is specific and/or different in cancer cells?
Reply: We deleted this part.
I am not sure that the section#3 (now section#4) about splicing and chromatin has been reduced as requested. I am still not convinced that this section is mandatory and again, I don’t see a clear connection with cancer in this chapter. What are the specific epigenetic mechanisms involved in splicing regulation in cancer?
Reply: We reduced the volume of this part and only described the content about cancer-related splicing and epigenetics. (section #4)
Section #7 : There are other spliced variants in other types of cancers (Met exon 14, BCL2, FAS ..). I propose to add a table with relevant examples. Please move the ARV7 paragraph in section #8. It would be more obvious.
Reply: Thank you for your kind comment. We added the other spliced variants in Table1. We also move the part about ARV7 to section #8.
Minor comments:
- U2 and U12 introns should be mentioned earlier
Reply: We added the comment about U2 and U12 introns. (page.2, lines.51-54)
- BRD9 itself is not the ncBAF complex. It is one specific member.
Reply: We revised this part. (page.5, lines.205-206)
- «are thought to be due to synthetic lethality caused by their functional alterations » Please can you clarify this point ?
Reply: We revised this point to make it easier to understand. (page.6, lines.234-235)
- characteristic: ? «: »is probably a typo Line 281
Reply: We revised this part. (page.7, line.241)
«are concentrated in P95 ». What is P ? I assume it is a proline residue at position 95 ? Same comments for S34 and Q157 ? Please change concentrate
Reply: We revised this part. (page.7, line.257, line.271, line.272)
«the isoforms thus produced lose their pro-apoptotic functions ». Please change the text. Line 428
Reply: We changed the text. (page.9, lines.330-331)
- Figure 1: This is a too schematic view of spliceosome activation and splicing regulation ; It does not only depends on U1. Please add the other U snRNAs in the figure
Reply: We added the other U snRNAs in Figure.1.
Round 3
Reviewer 2 Report
In Figure 1A. One step is missing. In fact both U1 and U4 are leaving the splicesome . Only U2/U5 and U6 are implicated in the catalytic spliceosome and regulate splicing reaction.
Please adapt the figure.